# The Effect of the National Plan (2010–2020) on the Development of Special Education in China: Evidence from Before–After Design at a 7-Year Interval

**Ahmed Alduais** [1,*] and Meng Deng [2]

1    Institute of International and Comparative Education, Beijing Normal University, No. 19 Xinjiekou Wai St, Beijing 100875, China

2    Institute of Special Education, Beijing Normal University, Xinjiekou Wai St, Beijing 100875, China; mdeng@bnu.edu.cn

*    Correspondence: ibnalduais@gmail.com

**Abstract:** The possible effect of the National Plan on the development of special education has not been examined, and there is no published evidence concerning both national and international readership about the realisation of this policy document in China. Given this, we conducted a before–after design study at a 7-year interval including six variables of special education: number of schools, total enrolment, new enrolment, graduates, educational personnel, and full-time teachers. The data were retrieved from the National Bureau of Statistics of China (NBSC). The results indicated two patterns of special education development in China. First, the National Plan has quantitatively affected some special education services (schools, new enrolments, educational personnel, and full-time teachers). Second, the National Plan has possibly resulted into better control of the quality of special education—evidenced by an insignificant increase in total enrolment and graduates at the two compared intervals.

**Keywords:** before-after design; census bureau data; China; national plan 2010–2020; special education development

## 1. Introduction

The National Plan for Medium and Long-Term Education Reform and Development (2010–2020) (hereafter the National Plan) was issued on July, 2010 [1] and "promulgated to guide and allocate tasks for the development of the education of the next decade" and "to promote the scientific development of education and propose the development goal of changing China from the world's largest education system to one of the world's best" [2] (p. 39). The execution of the National Plan has been challenged by some researchers. For instance, Pan [3] compared the planned reforms on this new policy with the past reforms in China—asserting that the education conflict has been the execution of the enforced policies rather than the enforcement of new ones. For the present study, we wanted to identity the given importance for special education in the National Plan. First, we used the Automatic Recognition of Multi-Word Terms (http://www.nactem.ac.uk/software/termine/) to identity the c-value [4] for any special education related terms. Four terms were generated based on the analysis of the National Plan. These included special education (rank = 7 and c-value = 17.66), special education system (rank = 73 and c-value = 3.17), existing special education (rank = 250 and c-value = 1.58), and special education development (rank = 250 and c-value = 1.58). We should note that the c-value is an online software that is used to calculate the frequency of a certain concept. Since we wanted to measure the importance of the special education field in the National Plan document, we used this software to check the given

attention to this field based on this document [4]. Second, we checked the contents page to see if there was a specific chapter and/or section for special education. The result indicated the inclusion of one chapter, namely, chapter 10 for special education development within the National Plan. It included three acts (28–30): (1) caring for and supporting special education, (2) improving special education system, and (3) perfecting guarantees for special education [5] (p. 3).

## 2. Literature Review

Several studies have reported the efforts of the Chinese government towards providing access to an improved quality of education to all. Among these efforts are those issuing regulations enforcing free special education for all children with disabilities and promoting a zero rejection strategy [6,7]. Pang [8] reviewed education laws in China. The reviewed educational laws included: Compulsory Education Law (CEL 1986), Compulsory Education Law-Revised (2006), Education Law for the Disabled (1994), People's Republic Law for the Disabled (1990), Law for the Disabled (1994), Regulations on the Vocational Opportunities for the Disabled (2007), and Development Guideline for the Disabled (2006–2010). According to Malinen, Savolainen, and Xu [9], the development of special education could be affected by the attitude of teachers themselves, particularly the tendency to uphold inclusive education. They examined this—using the Teacher Efficacy for Inclusive Practices scale (TEIP), which included three variables: efficacy of inclusive education, efficacy in collaboration, and efficacy in managing behavior. Based on this, the authors believe "future pre- and in-service teacher education programs should emphasize developing teachers' self-efficacy, particularly their collaboration skills" (p. 532). Other researchers approach special education in China in terms of identifying areas for improvement. For example, Ellsworth and Zhang [10] emphasise the need to develop special education services in China in different areas, such as providing full access to available information regarding special education, better teacher training (this was examined in-depth by Yu, Su, and Liu [11] who state the poor quality of special education teacher education and lack of national certification is hindering improvement of national special education services), international collaboration, increased funding, and more special education undergraduate and graduate programmes. These points were also maintained by Mcloughlin, Zhou, and Clark [12], who link the need for such development to the economic growth and prosperity of China-'the PRC [People's Republic of China] is at the cusp of a new era due to social forces that will make an indelible mark on the country's future" (p. 273). Furthermore, inconsistent classification and/or definition of the special education population between China and international organisations such as the World Health Organisation (WHO) is among the factors hindering special education development in China. Malinen et al. [13] considered this comparatively in his paper. Inclusion criteria in China sinclude visual, hearing, language, intellectual, and physical and mental disabilities, compared to cognition, mobility, self-care, getting along, life activities and participation, according to the WHO [13]. To reach a standard model of special education services in China seems to be a matter of time to some academics. For instance, Trube, Li, and Chi [14] conclude their qualitative paper on early childhood special education in China with the statement, "education for all and education according to one's needs and potentials is congruent with Chinese philosophy . . . the country has made significant progress in some areas of special education" (p. 112). It should be noted also that special education services in China have levels that differ from one province to another. These differences are usually affected by population density and the strategic plan of each province (Holdsworth, as cited in [7]). Additionally, the limitation of special education services is also an area that needs to be addressed by the government according to Hu and Szente [15]. They stated that special education services have been limited to certain types of disabilities, i.e., mental retardation and visual and hearing impairment, thus disregarding other disabilities such as emotional disorders and speech-language disorders. This is emphasised by Huang, Jia, and Wheeler [16], who state "since the late 1970s, special education in the People's Republic of China has experienced significant reform and fast development . . . education for children with severe developmental disabilities, especially autism spectrum disorders (ASDs), is still the greatest challenge in the field' (p. 1991). This indicates

that the government has possibly succeeded in developing special education at the macro level but not micro level (i.e., we are using the macro-level to indicate general special education services like having schools for special education and the micro-level to indicate that special education services are moving toward more specific services where each and every class of special education is fully considered and provided with learning materials according to each learner's special needs). Lytle, Johnson, and Yang [17] argued in favour of these micro and macro levels of special education where, according to them, government development and implementation of deaf education programs is urgently needed. They maintain that deaf people have " . . . limited higher education opportunities . . . no support services . . . no deaf teacher preparation . . . jobs are few" (p. 457).

## 3. The Present Study

This study examines the possible effect of the National Plan on special education development before and after the enforcement of this policy at a 7-year interval through census bureau data retrieved from the NBSC database. The following hypothesis guides this study:

$H_0$: There will be no statistically significant difference between the total number of special education schools, total enrolment, new enrolment, graduates, educational personnel, and full-time teachers before and after the National Plan at a 7-year interval;

$H_A$: There will be statistically significant difference between the total number of special education schools, total enrolment, new enrolment, graduates, educational personnel, and full-time teachers before and after the National Plan at a 7-year interval.

## 4. Method

### 4.1. Sample

The population of interest in our study is learners with special educational needs. However, we used the database available on NBSC database at a 7-year interval before and after the execution of the National Plan. The pre-intervention period is years between 2004 and 2010 and the post-intervention period is the years between 2011 and 2017. Yet, it should be noted that the data of every year present the data for the previous year. For instance, the data for 2017 in the database are the data for 2016 and so on. The data are presented on the website as chapters related to each educational field. For instance, the special education has one section, and the data present the number of special education schools, total enrolment, new enrolment, graduates, educational personnel, and full-time teachers. The data also included educational statistics for female versus male, but it was missing for some years so we decided to exclude the gender variable from the study. With this in mind, we propose that our results are only indicators of the level of development of special education and whether the National Plan has impacted this development or not; thus, they cannot be generalised and/or used to refer to the overall quality of special education as our tested hypothesis is purely quantitative. Furthermore, direct contact with the special education environment did not take place. We are also not including other factors like population or regular education system.

### 4.2. Measures

This study used unobtrusive measures. Secondary analysis of census bureau data was the main approach for examining our proposed hypothesis regarding special education development in China before and after the execution of the National Plan. To examine the development of special education services in China, there is a need to compare the services before and after the year 2010 (the year when the 2010–2020 National Plan was put into effect). An interval of 7 years before and after 2010 is used to examine such development. This helps one to understand and evaluate whether this development is due to this plan or not. That said, this study benefited from secondary data conducting a comparison between two intervals (7 years before 2010 and 7 years after 2010). The raw data were presented in

terms of numbers of special education schools, total enrolment, new enrolment, graduates, educational personnel and full-time teachers. The National Plan was executed in 2010, and the data include 7-year interval before and 7-year interval after. The National Plan represents the intervention and is the independent variable including data of either the pre-intervention period or the post-intervention period. The dependent variables are the aforementioned six variables of special education.

*4.3. Design*

A quasi-experimental design, namely before and after design was used in this study. In notational form, it can be depicted as follows:

$$O \longrightarrow X \longrightarrow O$$

where

X = the National Plan
O = the special education variables before and after the intervention at a 7-year interval.

The main assumption for this design is that the National Plan has affected the development of special education in China. For this reason, and as quantitative indicators, the number of special education schools, total enrolment, new enrolment, graduates, educational personnel, and full-time teachers will be statistically yet significantly higher in the post-intervention period than the pre-intervention period.

*4.4. Procedure*

The data were retrieved on September 10, 2018 from the NBSC database (http://www.stats.gov.cn/english/Statisticaldata/AnnualData/). They include census bureau data for all sectors in China between 1999 and 2017. Data from the census are presented in tables, reflecting the total number of special education schools, total enrolment, new enrolment, graduates, educational personnel, and full-time teachers. The data was converted into excel file and then analysed using the SPSS version 20. The independent samples *t*-test was mainly used to test the hypothesis in relation to the possible effect of the National Plan on the development of special education before and after execution at a 7-year interval.

**5. Results**

Tables 1 and 2 show the results of the independent-samples *t*-test, which was conducted to compare the number of the special education schools, total enrolment, new enrolment, graduates, educational personnel, and full-time teachers before and after the execution of the National Plan 2010–2020 at a 7-year interval, respectively. While Table 1 presents the descriptive statistics analysis, Table 2 shows the results of the inferential statistics.

We first ran descriptive statistics for the collected data to make an overall presentation about the possible differences among the examined variables before and after intervention of the National Plan. Table 1 includes five illustrative columns for the minimum, maximum, means, standard deviations, and standard deviation errors for the examined variables. As is seen, all the minimum values are less during the pre-intervention period than those in the post-intervention period. This output continues in the maximum values column, except in the graduates where the pre-intervention exceeds the post-intervention one. This is also reflected in the means, standard deviations, and standard deviation errors. Since it is not possible to decide on the significance among these recorded differences, Table 2 shows the significant and/or insignificant differences between the pre-intervention and post-intervention periods as explained below.

**Table 1.** *Descriptive statistics* for special education variables before and after the National Plan at a 7-year interval.

| Variable | Intervention | N | Minimum | Maximum | M | SD | Std. Error Mean |
|---|---|---|---|---|---|---|---|
| Special education schools | Pre-Intervention | 7 | 1551 | 1672 | 1605.57 | 42.751 | 16.158 |
| | Post-Intervention | 7 | 1706 | 2080 | 1913.14 | 143.308 | 54.165 |
| Total enrolment | Pre-Intervention | 7 | 362,946 | 428,125 | 389,864.14 | 30,023.194 | 11,347.701 |
| | Post-Intervention | 7 | 368,103 | 491,740 | 414,290.86 | 42,665.927 | 16,126.205 |
| New enrolment | Pre-Intervention | 7 | 49,000 | 64,018 | 55,535.43 | 7283.912 | 2753.060 |
| | Post-Intervention | 7 | 64,086 | 91,521 | 72,311.29 | 10,797.498 | 4081.071 |
| Graduates | Pre-Intervention | 7 | 43,214 | 64,018 | 50,159.29 | 6859.745 | 2592.740 |
| | Post-Intervention | 7 | 44,194 | 59,164 | 51,937.00 | 5526.326 | 2088.755 |
| Educational personnel | Pre-Intervention | 7 | 41,000 | 47,466 | 43,790.00 | 2431.590 | 919.055 |
| | Post-Intervention | 7 | 49,249 | 62,468 | 55,503.57 | 4650.717 | 1757.806 |
| Full-time teachers | Pre-Intervention | 7 | 30,000 | 37,945 | 33,661.71 | 2896.430 | 1094.748 |
| | Post-Intervention | 7 | 39,650 | 53,213 | 45,997.57 | 4882.963 | 1845.586 |

**Table 2.** *t*-Test results for special education variables before and after the National Plan at a 7-year interval.

| Special Education: | Equal Variances: | Levene's Test | | *t*-Test for Equality of Means | | | | | | | |
|---|---|---|---|---|---|---|---|---|---|---|
| | | F | Sig. | t | df | Sig. (2-Tailed) | Mean Difference | Std. Error Difference | 95% Confidence Interval Difference | | |
| | | | | | | | | | Lower | Upper |
| Schools | assumed | 10.583 | 0.007 | −5.441 | 12 | 0.000 | −307.571 | 56.524 | −430.727 | −184.416 |
| | not assumed | | | −5.441 | 7.060 | 0.001 | −307.571 | 56.524 | −441.001 | −174.141 |
| Total enrolment | assumed | 0.468 | 0.507 | −1.239 | 12 | 0.239 | −24,426.714 | 19,718.641 | −67,389.942 | 18,536.513 |
| | not assumed | | | −1.239 | 10.772 | 0.242 | −24,426.714 | 19,718.641 | −67,939.509 | 19,086.080 |
| New enrolment | assumed | 0.888 | 0.365 | −3.408 | 12 | 0.005 | −16,775.857 | 4922.853 | −27,501.831 | −6049.883 |
| | not assumed | | | −3.408 | 10.524 | 0.006 | −16,775.857 | 4922.853 | −27,671.074 | −5880.640 |
| Graduates | assumed | 0.008 | 0.930 | −0.534 | 12 | 0.603 | −1777.714 | 3329.444 | −9031.950 | 5476.521 |
| | not assumed | | | −0.534 | 11.480 | 0.604 | −1777.714 | 3329.444 | −9068.566 | 5513.137 |
| Educational personnel | assumed | 2.786 | 0.121 | −5.905 | 12 | 0.000 | −11,713.571 | 1983.568 | −16,035.395 | −7391.748 |
| | not assumed | | | −5.905 | 9.052 | 0.000 | −11,713.571 | 1983.568 | −16,196.767 | −7230.376 |
| Full-time teachers | assumed | 2.122 | 0.171 | −5.749 | 12 | .000 | −12,335.857 | 2145.848 | −17,011.257 | −7660.457 |
| | not assumed | | | −5.749 | 9.757 | .000 | −12,335.857 | 2145.848 | −17,133.287 | −7538.427 |

To start with the number of the special education schools, there was a significant difference between the schools' number for the pre-intervention (M = 1605.57, SD = 42.75) and post-intervention (M = 1913.14, SD = 143.31) periods; $t$ (12) = −5.44, $p < 0.001$. These results suggest that the National Plan has an effect on special education development. Specifically, our results suggest that since this new policy was started, the number of special education schools has been increasing.

The second dependent variable was total enrolment, and there was not a significant difference between the total enrolment for the pre-intervention (M = 389,864.14, SD = 30,023.19) and post-intervention (M = 414,290.86, SD = 42,665.92) periods; $t$ (12) = −1.24, $p = 0.24$. These results indicate that the National Plan does not have an effect on the total enrolment of students in special education. Particularly, our results suggest that between the execution of the national plan and now, the enrolment level has not significantly increased.

The third dependent variable was new enrolment, and there was a significant difference between the new enrolments for the pre-intervention (M = 55,535.43, SD = 7283.91) and post-intervention (M = 72,311.29, SD = 10,797.50) periods; $t$ (12) = −3.41, $p < 0.005$. These results predict that the National Plan has an effect on new enrolment levels. Peculiarly, our results suggest that since the new policy was executed, the number of newly enrolled students in special education has been increasing significantly.

The fourth dependent variable was graduates, and there was not a significant difference between graduates for the pre-intervention (M = 50,159.29, SD = 6859.745) and post-intervention (M = 51,937.00, SD = 5526.32) periods; $t$ (12) = −0.53, $p = 0.60$. These results assume that the national plan does not have an effect on the number of graduates of special education. Our results suggest that since the implementation of the National Plan, the number of graduates has not significantly increased.

The fifth dependent variable was educational personnel, where there was a significant difference in the educational personnel for the pre-intervention (M = 43,790.00, SD = 2431.60) and post-intervention (M = 55,503.57, SD = 4650.72) periods; $t$ (12) = −5.90, $p < 0.001$. These results put forward that the National Plan does have an effect on the number of educational personnel of special education. Explicitly, our results suggest that the implementation of the new policy has resulted in higher demand for educational personnel in special education.

The last dependent variable was full-time teachers, and there was a significant difference between the full-time teachers for the pre-intervention (M = 33,661.71, SD = 2896.43) and post-intervention (M = 45,997.57, SD = 4882.96) periods; $t$ (12) = −5.75, $p < 0.001$. These results imply that the national plan has an effect on full-time teachers of special education. Obviously, such results suggest that the National Plan increased the demand for full-time teachers of special education.

## 6. Discussions

The results of this study were not completely in agreement with the initial expectations. The alternative hypothesis proposed that there would be statistically significant difference between the total number of special education schools, total enrolment, new enrolment, graduates, educational personnel, and full-time teachers, before and after the National Plan at a 7-year interval. While four variables (i.e., schools, new enrolment, educational personnel, and full-time teachers) credited the alternative hypothesis, two variables (i.e., total enrolment and graduates) discredited it. Put differently, although there were minor differences in means and standard deviations in the two variables between the two intervals (before and after), such differences were not statistically significant.

There are two major possible interpretations for this outcome pattern. First, if we accept the assumption evidenced in our study that the numbers of special education schools, new enrolments, educational personnel, and full-time teachers have significantly increased due to both the enforcement and the execution of the National Plan 2010–2020, this would imply that such policy has resulted in the development of special education in China, yet such policy has not been realised. On the other hand, we could assume that the increase of numbers in these variables that were measured at a 7-year interval, before and after, can be attributed to other factors. For instance, the increase could be due to population growth, especially after the two-child policy enforcement. Or, it could be due to inefficient

and ineffective prevention, provision, and treatment methods in the area of special education, resulting in high demand for special education provision. In this regard, prior research reported the need for further training for regular education teachers towards more inclusive education [18–20] (limitations of Learning in Regular Classrooms (LRC) [21–24]). Second, it is very interesting to see the number of total enrolment and graduates before and after the National Plan close to one another. Generally, the simple indication states that there are relatively similar enrolment and graduation rates between the two intervals—leading to the conclusion that such policy was inefficient and ineffective in terms of these two variables. However, the new enrolment variable suggests that the decrease in total enrolment can be attributed to a policy to control both the enrolment and graduation rates, especially when linking this to inclusive education and mainstream education to ensure the better quality of special education services that are provided. Although this inference is not evidenced in our study, the National Plan, which stated clearly the development of special education both quantitatively and qualitatively, could roughly support this inference. These issues are also implied in previous studies, including [25]—approaching administrative concerns; [26]—raising students' related concerns; and [27]—raising the issue of the placement of special education students via regular education teachers. The model below illustrates the state of special education patterns in China based on the above-discussed evidence.

To summarize the status of special education based on this quantitative synthesis, the authors attempted the model below (Figure 1). There seems to be three controlling patterns of special education in China, namely, context, change, and time-scale. The Chinese context for special education and conflict between longing for the Chinese tradition and the desire for western traditions has established the existing conflict and entire development of this field of education. Hence, the driver of change will start from the government attempts to reform and level up special education in particular and education in general. This is to be achieved within a time-scale as determined in the National Plan 2010–2020. Expansion efforts have been led by the National Plan, which accounts for population size and recently decreed (in 2016) a two-child policy. However, since these three factors are more focused on quantity, enrolment and graduation control can roughly ensure quality control and balanced development of special education. In spite of this noticeable change, expansion of special education by itself is an indirect call and promotion for segregation that goes against the international efforts toward inclusion of learners with special needs. This also raises the conflict and concern of both policy makers and educators about the impact of special education on regular education, should complete inclusion be enforced. For this reason, implementing segregation for certain cases, mainstreaming for possible cases, inclusion for mild cases, and exclusion for some other situations seems to be the current model of special education in China.

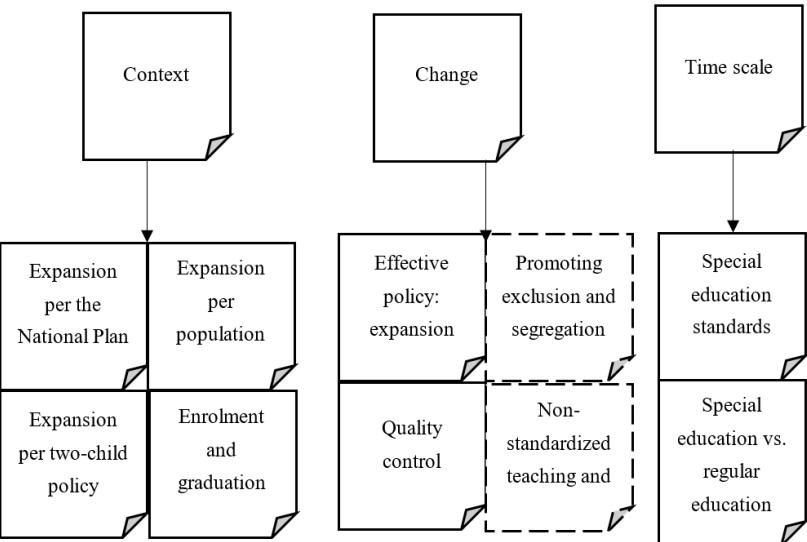

**Figure 1.** A model for the development and status of special education development.

## 7. Conclusions and Recommendation

To examine the effect of the National Plan 2010–2020 on the development of special education in China, a before–after design study at a 7-year interval was conducted. The data, which were retrieved from the NBSC database, included six variables as measuring indicators for the development of special education. After testing the generated data for 7 years before and 7 years after the enforcement of the National Plan, the results indicated two patterns of this investigated topic. First, the enforcement and execution of this policy has possibly but not definitely contributed to the development of special education, as evidenced by the recorded increase in special education schools, new enrolment, educational personnel, and full-time teachers. These four factors represent the greater provision services of special education. Second, the same policy has not resulted in the better development of enrolment and graduation rates in the special education community, albeit, this was interpreted from our perspective as control of quality rather than a drawback in this policy. In other words, the presented data and results did not indicate a significant increase in the numbers of enrolment and mainly graduates, hindering the development process, which we attribute to the [efforts] of the government to control the quality of the introduced special education services. Despite this, the presented data are insufficient to support the claim that the absence of increase in enrolment and graduates is due to the intention of the government to control the quality of special education. That is to say, this could be either a shortcoming of provisions (i.e., the government has been able to expand special education services but it has somehow failed to match the curriculum and learning environment to the level of the learners, resulting in lower levels of enrolment and, more importantly, fewer graduates) or a positive factor supporting better treatment methods resulting in controlled enrolment and graduation rates (i.e., implementing better standards for both enrolment and graduation as an effective provision method, resulting in better special education quality).

The results of this study have at least three implications: epistemological, educational, and managerial. Epistemologically, although the presented evidence showed a minor effect of the examined policy on the special education variable before and after the intervention, the evidence is abstract and biased towards quantitative-based evidence. The recorded differences, even if they are statistically significant, were biased towards the educational statistics variables. It would have been more effective if other variables like population growth and regular school education enrolment had been examined to obtain more concrete evidence to decide if the increase in the post-intervention was really due to the National Plan 2010–2020. Second, there seems to be a conflict in the prevention, provision, and treatment methods in the special education system in China. This was reflected through the variable results, especially in the new enrolment, compared to the total enrolment and graduates. If new enrolments are increasing, this means there is an increase in the size of the special education population—indicating a problem that needs to be solved by searching for why the demand for special education is increasing in the first place, other than increasing its provision. If there is a need to increase something, it should be the quality of the current services of special education. Third, the National Plan did not state exactly how the development of special education will be realised by 2020, it merely stated theoretical objectives to achieve a better special education system including, for instance, that every county with over 300,000 people and a large number of students with disabilities should have a special education school. Hence, unlike this forward-looking perspective of special education development in China, prior research has challenged the quality of this development (e.g., [28–30]).

**Author Contributions:** Conceptualization and project administration was handled by the second author. The first author was responsible for methodology, formal analysis, data collection and writing up of the paper including submission of the paper for publication.

**Funding:** This research received no external funding.

**Conflicts of Interest:** There is no conflict of interest.

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
