# Peer review of "The Effect of the National Plan (2010–2020) on the Development of Special Education in China: Evidence from Before–After Design at a 7-Year Interval"

_education, doi:10.3390/educsci9020095_

Round 1
Reviewer 1 Report
I am not sure that the methods used are appropriate to make the case of this article. Taking a 7 year period before the National Plan and the next years for after the National Plan makes it difficult to measure success. I think a smaller window maybe years 5-7 after the National Plan and 3 years before the national plan may be better. I also think that since the National Plan impacts counties larger than 300,000 population that these are the counties that should be looked at for the study. Maybe a comparision before and after in the 300,000+ counties with the before and after <300,000 could yeild better data for this study.
I have made comments within the document highlighting other concerns.

Author Response
Dear reviewer,
Thank you so much for your efforts and valuable comments towards improving our paper.
Please see below our reply to the comments and attached is the modified version of the paper.
We do really thank you for your efforts.
We ran the spelling checker again to verify and go through the typos.
The study design has been elaborated and addition are coloured in blue.
The selection of 7-year interval was based on several recommendation after some consultation with some professors and statisticians. It was also recommended by a reviewer of an earlier version of the paper based on a study of 5 years after the exaction of the plan who suggested that there must be some years before and some years after. The main reason for choosing 7 years in particular is based on the availability of the data. Especially the data after 2010. When the paper was conducted, the only accessible data was for 7 years after 2010. As for the suggestion based on counties, we think this would change the focus of the study where the deciding factor would be the expansion of special education and fair distribution of special education services based on population size. We do think, this could make an interesting future study which seems to be beyond the objectives of our current study.
The other provided comments highlighted in yellow by the reviewer have been accounted for and modified.
Second, the National Plan has possibly resulted into a better control of the quality of special education—evidenced through insignificant increase in the total enrolment and graduates at the two compared intervals.
From our end, we inferred this through the analysis of the total enrolment and graduates and we consider this as an implication other than an output.
We add some lines about the functions of c-value.
Holdsworth is an author and we are using indirect citation as per the APA style.
Johnson argued that...(as cited in Smith, 2003, p. 102).
https://owl.purdue.edu/owl/research_and_citation/apa_style/apa_formatting_and_style_guide/in_text_citations_author_authors.html
For the selection of 7 years, it was based on this feedback provided by a prior reviewer:
To examine the development of special education services in China, there is a need to compare the services before and after the year 2010 (the year when the 2010-2020 national plan was put into effect). An interval of 6 or 7 years before and after 2010 is important for such studies to examine such development. This will help to understand and evaluate whether this development is due to this plan or not. The study could benefit more from other secondary data as indicated above. I recommend a comparison between two intervals (7 years before 2010 and 7 years after 2010).
The increase of the special education personnel is an evidence by itself and we stated that this is just an implication based on the collected and analysed secondary data. Besides, a major objective of the National Plan was expansion of special education of which increase of educational personnel.
We elaborated on the presented model at the end of the discussion section.
Reviewer 2 Report
The tables within the article are very good and in depth. However, the synopsis should be simplified somewhat. It should be very succinct and easy for the reader who does not have such an in depth background in research. More application of the tables should be discussed for the reader.
The literature review was excellent and very in depth with current literature. This helped the reader with a better understanding of the article.
The abstract was good. It created a basic understanding of the article.
Author Response
Dear reviewer,
Thanks so much for your comments and efforts towards improving our paper.
We worked on some modifications as per suggested by you and the other reviewers.
We are attaching the paper for your reference, Additions are coloured in blue.
Once again, thanks and wishing you a good day.
Regards,
Round 2
Reviewer 1 Report
Many corrections were not made. The expansion of the text did not clarify my concerns. APA citations within the text were not corrected.
Author Response
Dear Reviewer and editors,
Thanks for your second round feedback.
We modified the paper citation and references based on the provided comments and provided sample.
As for other comments, we went through comments of round 1 again, and we do think that we have covered these on our earlier version. Should there be any more specific comments that need to be approached, we would be happy to work on them.
We are attaching the modified version here. Changes are all made using the Word Review Function.
Regards,
